# Development of an Easy-to-Use Prediction Equation for Body Fat Percentage Based on BMI in Overweight and Obese Lebanese Adults

**DOI:** 10.3390/diagnostics10090728

**Published:** 2020-09-21

**Authors:** Leila Itani, Hana Tannir, Dana El Masri, Dima Kreidieh, Marwan El Ghoch

**Affiliations:** Department of Nutrition and Dietetics, Faculty of Health Sciences, Beirut Arab University, P.O. Box 11-5020 Riad El Solh, Beirut 11072809, Lebanon; l.itani@bau.edu.lb (L.I.); hana.tannir@bau.edu.lb (H.T.); dana.masri@bau.edu.lb (D.E.M.); d.kraydeyeh@bau.edu.lb (D.K.)

**Keywords:** bioimpedance, body composition, body fat percentage, obesity, predictive equation

## Abstract

An accurate estimation of body fat percentage (BF%) in patients who are overweight or obese is of clinical importance. In this study, we aimed to develop an easy-to-use BF% predictive equation based on body mass index (BMI) suitable for individuals in this population. A simplified prediction equation was developed and evaluated for validity using anthropometric measurements from 375 adults of both genders who were overweight or obese. Measurements were taken in the outpatient clinic of the Department of Nutrition and Dietetics at Beirut Arab University (Lebanon). A total of 238 participants were used for model building (training sample) and another 137 participants were used for evaluating validity (validation sample). The final predicted model included BMI and sex, with non-significant prediction bias in BF% of −0.017 ± 3.86% (*p* = 0.946, Cohen’s d = 0.004). Moreover, a Pearson’s correlation between measured and predicted BF% was strongly significant (r = 0.84, *p* < 0.05). We are presenting a model that accurately predicted BF% in 61% of the validation sample with an absolute percent error less than 10% and non-significant prediction bias (−0.028 ± 4.67%). We suggest the following equations: BF% _females_ = 0.624 × BMI + 21.835 and BF% _males_ = 1.050 × BMI − 4.001 for accurate BF% estimation in patients who are overweight or obese in a clinical setting in Lebanon.

## 1. Introduction

According to the World Health Organization (WHO) [1] on a global scale, more than 1.9 billion adults are overweight (body mass index (BMI) ≥ 25 kg/m^2^), and of these over 650 million are obese (BMI ≥ 30 kg/m^2^) [1]. In other words, nearly 39% of adults were overweight in 2016, and 13% were obese [1]. Moreover, according to the same source, since 1975 the prevalence of obesity in the world has nearly tripled [1]. Beyond the BMI cut-off, the WHO originally defines overweight or obesity as an increased body fat (BF) accumulation in the adipose tissue, especially in central visceral regions [1,2]. Thus, the classification of adiposity based on BF quantification and assessment seems to be the most accurate [3], especially when, regardless of BMI, an increase in BF% is associated with major risk factors for cardiovascular diseases [4] as well as all-cause mortality [5]. Therefore, an accurate measurement and/or estimation of the BF% in patients with obesity are of clinical importance [6].

BF can be measured by a wide range of techniques which vary in the level of precision and accuracy. On the one hand, techniques such as magnetic resonance imaging (MRI), air displacement plethysmography, computed tomography (CT), and dual-energy X-ray absorptiometry (DXA) [7] are more likely to be used in a research setting since they require trained technicians and/or involve sophisticated costly instrumentation [8,9]. On the other hand, in recent years, bioelectrical impedance analysis (BIA) devices have become one of the most popular and convenient ways to measure BF% and body composition and have been widely used in clinical practice and epidemiological studies. BIA is a non-invasive, simple-to-perform, and time-saving technique and shows good levels of accuracy and agreement with gold-standard assessments (e.g., DXA) in several weight management settings with overweight and obese individuals [10,11,12]. Although concordance between BIA and DXA methods at the individual level is lacking, both methods can be used interchangeably at a population level [13]. However, BIA is not regularly available in all clinical settings, especially in single-therapist outpatient clinics. The devices are still considered relatively expensive and can cost tens of thousands of dollars, whereas simple models that may be used in individual clinical practice are much more affordable [14]. 

To this aim, predictive equations based on anthropometric measurements (i.e., body weight, height, age, and sex), can be used to estimate BF%, as shown by Womersley and Durnin [15], Jackson [16], Deurenberg [17], Gallagher [18], and Gomez Ambrosi [19]. However, in our previous study we showed that the majority of these equations are unsuitable for accurately estimating BF% among our population, i.e., Lebanese patients who are overweight or obese. This is because some of these equations were originally developed and validated in individuals from the general population (i.e., not a clinical sample and only involving individuals with standard weight status) and/or in certain populations (i.e., Western society) [15,16,17,18]. Therefore, it is uncertain if they are valid among other populations, i.e., patients with obesity and of a different ethnicity [20]. Finally, to the best of our knowledge, no BF% predictive equation has been developed for overweight or obese individuals who are from Arabic-speaking countries and, specifically, who are Lebanese.

These considerations prompted us to focus the present study on developing simplified BF% prediction equations for a clinical sample of overweight or obese Lebanese adults using easily obtained anthropometric measurements that did not require specialized instruments. The study includes evaluating the validity and accuracy of the equations.

## 2. Materials and Methods 

Care was taken to adhere to the Transparent Reporting of a multivariable prediction model for Individual Prognosis Or Diagnosis (TRIPOD) guidelines [21]. Briefly, the TRIPOD is composed of a 22-item checklist to improve reporting studies that aim to develop, validate, and even update prediction models of a diagnostic or prognostic nature [21]. Therefore, the TRIPOD statement robustly increases transparency of the report of a prediction model [21].

### 2.1. Study Design

A single-measure, cross-sectional study was conducted.

### 2.2. Data Source and Study Population

A simplified prediction equation to predict BF% was developed and evaluated for validity in a clinical sample of both genders with overweight or obese individuals who were seeking treatment at the outpatient clinic of the Department of Nutrition and Dietetics at Beirut Arab University (Lebanon). Patients were recruited between September 2017 and March 2020. The inclusion criteria stipulated that patients should be aged ≥ 18 years, with a body mass index (BMI) of ≥ 25.0 kg/m^2^ with at least one weight loss responsive comorbidity (e.g., type 2 diabetes, cardiovascular disease, sleep apnea, severe joint disease, or two or more risk factors) as defined by the Adult Treatment Panel III [22]. Conversely, pregnancy or lactation, taking medication known to influence body weight or composition, or any clinical condition that did not indicate weight loss were considered as exclusion criteria. Participants were supposed to be enrolled in a program featuring a low-calorie diet. The protocol for treatment essentially involved a personalized cognitive behavioral treatment (CBT-OB) program designed for patients with obesity as described elsewhere. Patients were instructed to follow a low-energy diet (LED: 1200 kcal/day for females and 1500 kcal/day for males). The eating plan was based on the Mediterranean diet [23,24]. 

Through a baseline assessment, a dataset of 375 adult patients (with both BIA and anthropometric measurements) was randomly split into one group of 238 participants (177 females and 61 males) who were used to develop the prediction equation (training sample), and a second group of 137 patients (100 females and 37 males) who were used to evaluate the validity of the developed prediction equation (validation sample). 

### 2.3. Ethics Approval

The study protocol and procedures were conducted according to the Declaration of Helsinki and approved by the Institutional Review Board of Beirut Arab University (approval number: 2017H-0034-HS-R-0241, approved on 10 January 2017). All participants gave informed and written consent. 

### 2.4. Anthropometric Measurements

Participants were weighed (barefoot and wearing light indoor clothing) to the nearest 0.1 kg using an electronic weighing scale (SECA 2730-ASTRA, Hamburg, Germany). Their height was measured to the nearest 0.5 cm using a stadiometer. BMI was calculated according to the standard formula of body weight measured in kilograms, divided by the square of the height in meters. Participants were then classified as overweight or obese according to the WHO classification [1]. 

A multi-frequency segmental body composition analyzer (MC-780MA, Tanita Corp., Tokyo, Japan) was used to measure body composition based on three frequencies providing highly accurate, whole-body, and segmental measurements [25]. Participant age, gender, and height information were entered into the device [25]. Then, the participant was instructed to stand in a stable, barefoot position. Separate readings for different body segment compositions were obtained. These readings were based on an algorithm incorporating impedance, age, and height to estimate total BF, fat-free mass (FFM), and total body water (TBW) [25]. 

All recommendations for a correct BIA measurement were followed, such as taking measurements more than 3 h after waking, urinating before the measurement, no food or drink for at least 8 h beforehand, no heavy exercise during the 12 h prior to measurement, no alcohol or energy drinks in the 12 h beforehand, no metal objects, and no pacemakers. According to the literature, specifically this BIA model (MC-780MA, Tanita Corp., Tokyo, Japan) when compared with DXA used in other published studies [26,27], showed excellent reproducibility in assessing overweight and obese individuals, especially for BF (Intraclass correlation coefficient (ICC) = 0.88) and BF% (ICC = 0.66) [26]. Additionally, DXA and BIA have shown high significant correlations in BF% measurements (r = 0.852, ICC = 0.84, and concordance coefficient = 0.844) in young adults regardless of their level of physical activity [27].

### 2.5. Statistical Analysis 

#### 2.5.1. Descriptive Statistics

Descriptive statistics are presented as mean and standard deviation (SD) for continuous variables and proportions for categorical variables. Student’s *t*-test was used for mean comparison and chi-squared test of independence was used for categorical variables. 

#### 2.5.2. Predictors to Be Included in the Model

Initially, the potential to predict BF% from potential predictors was examined by a scatter plot to detect a linear or quadratic relationship with BF%. Pearson’s correlation was calculated between the dependent variable and BMI. To confirm linearity, the CUSUM test was used and a positive linear association between BF% and BMI was confirmed with a CUSUM test *p* > 0.05. Interaction of sex with BMI was also tested using a scatter plot for BMI versus BF% across sex categories. When interaction was confirmed with different variation in BF% in males or females, an interaction term was added to the prediction model.

#### 2.5.3. Model Derivation

A prediction equation was developed using an ordinary least squares linear regression analysis with a stepwise backward approach using predictors that were found to be correlated significantly with BF%. BMI, age, and sex and their interaction terms were included in the initial model. The least significant predictors were removed if the p-value criterion was not met (*p* < 0.1).

#### 2.5.4. Evaluation of Model Performance and Internal Validity and External Validity

Model performance and selection were assessed based on its explanatory power (R^2^) and low standard error of estimation (SEE) and total error (TE) $\sqrt{\sum {\left(Y-Y_{1}\right)}^{2}/N}$ (where *Y* is the measured BF% and *Y*_1_ is the BF% obtained using the estimation equation) [28]. Regression of the measured BF% on the predicted BF% (predicted used as independent variable) and Pearson’s correlation coefficient were calculated. A regression line with a slope of one and an intercept of zero indicated accurate prediction with no bias. A regression line with a slope significantly deviating from one suggested that a unit change in BF% did not correspond to a unit change in predicted BF%. A *t*-test was used to confirm that the slope was not different from one. Furthermore, a Bland–Altman plot was used to assess agreement between measured and predicted BF%. A paired sample *t*-test was used to assess if the mean bias was different from zero and Cohen’s d effect size was calculated. The closer Cohen’s d was to zero, the smaller the mean bias. The limits of agreement were calculated for a 95% CI as the mean difference ± 1.96 × SD. The mean percent bias between measured and predicted was calculated ((predicted-measured) × 100/measured) and agreement was defined at ±10% of the measured BF%. A prediction between 90% and 110% was considered to be an accurate prediction. Values above or below 10% were considered to be over- or underpredictions, respectively. All statistical significance was set at *p* < 0.05. All statistical analysis was carried out using SPSS version 25.0 (IBM Corp. Armonk. NY, USA) and NCSS 12 (NCSS, LLC, Kaysville, UT, USA, ncss.com/software/ncss). 

After finalizing the model derivation with the training sample, the model was applied to the validation sample to test external validity. The TE, Pearson’s correlation coefficient, the calibration slope, and R^2^ for the regression of the measured BF% on the predicted BF% were used to assess the performance of the model in the validation sample. The Bland–Altman method was also used to assess agreement between measured and predicted BF% in the validation sample. 

## 3. Results

### 3.1. Characteristics of the Study Participants

The characteristics of participants in the training and validation samples are shown in Table 1. The mean age (35.77 ± 14.72 years vs. 35.18 ± 15.85 years) and BMI (33.38 ± 5.16 kg/m^2^ vs. 33.39 ± 5.45 kg/m^2^) did not differ in either the training or validation samples. Overweight (28.6% vs. 29.9%), obesity (71.4% vs. 70.0%), and sex distribution (74.4% females vs. 73.0% and 27.0% vs. 25.6% males) were also similar in training and validation samples, respectively (Table 1). 

### 3.2. Model Predictors

The scatter plots and correlation analysis between measured BF% and potential predictors including age, sex, and BMI are shown in Table 2 and Figure 1. Predictors that were significantly correlated with BF% were sex (0.635 vs. 0.560) and BMI (0.461 vs. 0.413) in the training and validations samples, respectively (Table 2). The CUSUM test for linearity indicated a linear association between significant predictors and BF% (*p* > 0.05). 

### 3.3. Derived Model 

The stepwise multiple regression analysis yielded two potential models. The first model included BMI, sex, and age and their interaction terms. Although this model had good explanatory power (Table 3), the model revealed a bias of 0.584 ±3.81% in the training sample and 0.425 ± 4.82% in the validation sample, which was significantly different from zero in the training sample (Table 4). Removing either age or its interaction term and running the backward regression yielded the second model with a significant decrease in bias by 0.601% and absolute % error by 0.25%, and a lower effect size and higher prediction accuracy (69.3% vs. 67.69% within ±10% criteria). Hence, the second model, which included BMI and sex and their interaction term and had the least bias, was selected. It had a good explanatory power, SEE, TE, and prediction accuracy (Table 3 and Table 4). The following simplified equations present the selected model in males and females:BF% _females_ = 0.624 × BMI + 21.835BF% _males_ = 1.050 × BMI − 4.001

### 3.4. Model Performamce

The final selected model met the validity criteria as shown in Table 3 for both the training and the validation samples. In the training sample, the model explained 70.9% of the variance in BF% with an SEE and a TE of 3.88% and 3.85% BF with a calibration slope that is not different from one and an intercept of zero (Figure 2, Table 3). The prediction bias was minimal (−0.017 ± 3.86) and did not vary significantly from zero, with a Cohen’s d very close to zero (d = 0.004, *p* = 0.946) (Table 4). Pearson’s correlation between measured and predicted BF% was significant (r = 0.84, *p* < 0.05). The Bland–Altman plot revealed a limit of agreement of −7.58% to 7.54%, indicating that 95% of the samples predicted by this model lay within 8% above or below their actual value (Table 4, Figure 3). 

The accuracy of performance of the presented model was evaluated in the validation sample. The regression of measured BF% on predicted values had significant slopes and intercepts and explained almost 60% (R^2^ = 0.588) of the variance in the predicted BF% with an SEE of 4.65 and a TE of 4.65. The slope was not significantly different from one (0.898, *p* = 0.121) with an intercept of 4.02% (Figure 2, Table 3) indicating a minimal bias. A non-significant prediction bias of -0.028 ± 4.67% was observed together with a mean percent bias of 1.49 ± 13.14% with a mean absolute percentage error less than 10% (9.85 ± 8.78%). Cohen’s d indicated that the mean bias was small in the validation sample (d = 0.006, *p* = 0.943) and very close to zero, and was similar to that in the training sample (d = 0.004, *p* = 0.946) (Table 4). Pearson’s correlation coefficient between predicted and measured values in the validation sample was r = 0.77 (*p* < 0.05). A Bland–Altman plot revealed a limit of agreement between 9.12% and −9.18% (Figure 3). With respect to the accuracy of prediction in terms of proportion of predicted values falling within ± 10% of measured BF%, while the prediction accuracy was 69.3% in the training sample, it was 60.6% in the validation sample indicating that 61% of the observations fell within the 10% limits of accuracy.

## 4. Discussion

The current study aimed to develop a simple and easy-to-use BF% prediction equation based on BMI. Treatment-seeking overweight and obese Lebanese adults were studied in an outpatient clinical setting. One major finding was revealed. 

### 4.1. Findings and Concordance with Previous Studies

A simplified BF% predictive equation was generated and evaluated for validity in a clinical sample of females and males who were overweight or obese. It is difficult to compare our findings with previous studies on similar populations, i.e., Lebanese or individuals from other Arabic-speaking countries, since to the best of our knowledge such studies have been lacking.

It is true that a clinical setting usually relies on the WHO BMI cut-off points to define overweight and obesity in adults (i.e., ≥ 25 kg/m^2^ and ≥ 30 kg/m^2^ respectively) [1]; this is because this method is considered a simple and cheap tool of assessment, in addition to its strong correlation with BF [17]. However, BMI classification has its limitations [29]. First, BMI is not able to discriminate BF from lean body mass. Second, this currently available BMI classification to determine adiposity was determined involving Caucasian participants, and its validity is not a certainty in others (i.e., the Middle East and North Africa (MENA) region). Furthermore, several reports have underlined the need for revisions of BMI cut-offs to define overweight and obesity for non-Western ethnic groups [30]. Therefore, in the interim, the direct or indirect quantification of BF becomes vital in a population such as ours (i.e., the MENA region), and here stems the importance of the scope of our study.

### 4.2. Potential Clinical Implications

Our findings have a clinical implication. The study provides a simple equation to accurately predict BF% in overweight and obese Lebanese adults in a clinical setting. This method does not require specialized expensive equipment (MRI, DXA, BIA) or skilled professionals trained to collect accurate measurements (e.g., skinfold thickness). Adhering to standard protocols is not always achievable in patients with obesity [8,31]. In addition, some of these techniques are time-consuming and involve radiation (e.g., DXA) [32]. The only equipment required for our proposed method is a scale and a stadiometer, both being available in any clinical setting. 

Moreover, our finding can be implemented in an e-Health Application freely downloaded on smartphones, tablets, and computers, which is easy for professionals to use (i.e., doctors, nutritionists, dieticians, etc.), with the primary goal of improving healthcare through the optimization of patient care and a reduction in costs [33].

### 4.3. Strengths and Limitations 

Our study has certain strengths. Most importantly, it is the first study to develop a predictive equation for estimating BF% based on BMI (easily obtainable parameter), in a large clinical sample of overweight or obese Lebanese adult patients of both genders who are representative of the usual weight-management seeking treatment samples [34]. In addition, the study was conducted in adherence to TRIPOD guidelines, considered a gold-standard method of developing prediction models. In contrast, the study has several limitations. First and foremost, data were obtained in a single outpatient unit, which means that external validation is required. Furthermore, our results must be interpreted with caution because they may not apply to obese patients treated in other clinical settings (i.e., inpatient or bariatric surgery settings) who have severe obesity and different body composition patterns. Finally, BF was assessed by means of BIA, and this should be considered a major limitation since BIA is still not accepted as the gold-standard method when compared with DXA or hydrostatic underwater weighing. However, multi-frequency BIA has been found to be a very accurate measurement and in agreement with DXA in healthy adults [35,36], and has been widely validated in overweight and obese individuals in several clinical settings [10,11,12,37,38,39]. 

### 4.4. New Directions and Areas for Future Research

The current study opens the way for future research in our region (i.e., the Middle East and North Africa (MENA) region). First, as our finding involved a sample from a single unit, the external validity of our equation must be replicated in overweight and obese patients being treated in other outpatient clinical settings, and/or treated with different therapeutic weight management modalities (drug therapy, bariatric surgery, inpatients, etc.). Second, validation and/or development studies of existing or new BF% estimating equations should be conducted in other Arab-speaking countries. Third, there is also a need to develop a valid equation that accurately estimates BF% in healthy individuals in the general population in Lebanon and other countries in the MENA region. Last, but not least, the identification of optimum cut-off points for BF% in predicting obesity-related cardiovascular diseases is vital, similarly to what has been done in Western [40] and Asian [41,42] populations; otherwise, the usefulness of this measurement aim remains on a par with that of BMI [43].

## 5. Conclusions

In our study, we provided a simple prediction equation of BF% in overweight or obese Lebanese adults. The benefit of this equation is that it uses simple measurements (i.e., weight and height) that are easily obtained by health professionals (e.g., medical doctors, nutritionists, dieticians), especially in clinical settings (e.g., outpatient clinics) where sophisticated instruments such as MRI, CT, DXA, and BIA are not available. 

## Figures and Tables

**Figure 1 diagnostics-10-00728-f001:**
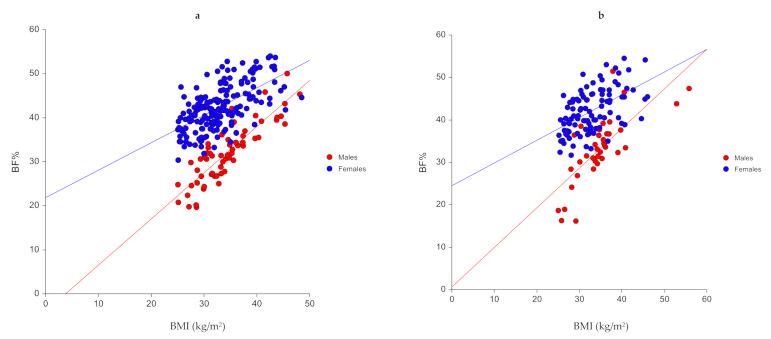
Scatter plots showing correlation between BF% and predictors in the (**a**) training sample (*n* = 238) and (**b**) validation (*n* = 137) samples.

**Figure 2 diagnostics-10-00728-f002:**
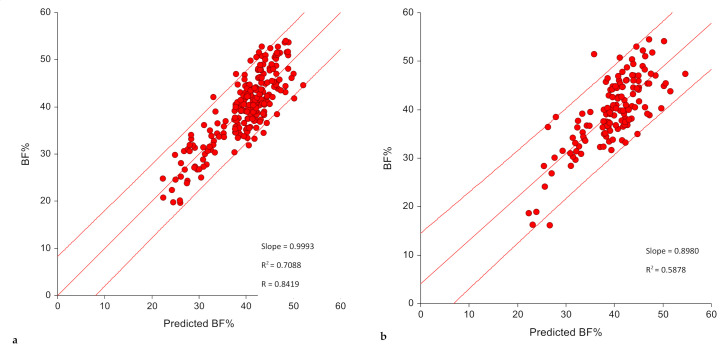
Calibration of prediction equation scatter plot of predicted vs. measured predicted BF% in the (**a**) training (*n* = 238) and (**b**) validation (*n* = 137) samples, model 2.

**Figure 3 diagnostics-10-00728-f003:**
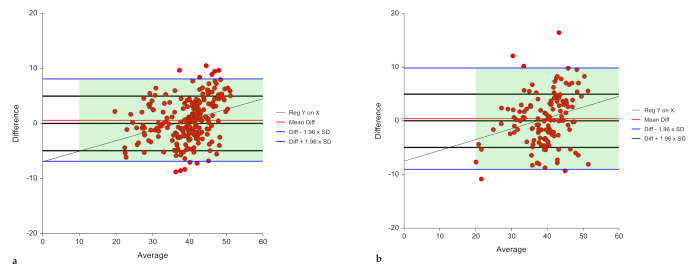
Bland–Altman plot for the agreement between measured and predicted BF% in the (**a**) training sample (*n* = 238) (**b**) validation (*n* = 137) samples, model 2.

**Table 1 diagnostics-10-00728-t001:** Descriptive statistics for training and validation samples.

	Total *n* = 375	Training Sample *n* = 238	Validation Sample *n* = 137	Significance
Sex	n (%)			χ^2^ = 0.085; *p* = 0.770
Males	98 (26.1)	61 (25.6)	37 (27.0)	
Females	277 (73.9)	177 (74.4)	100 (73.0)	
Age (Years)	35.55 (15.12)	35.77 (14.72)	35.18 (15.85)	*p* = 0.723
BMI (kg/m^2^)	33.39 (5.26)	33.38 (5.16)	33.39 (5.45)	*p* = 0.986
				χ^2^ = 0.077; *p* = 0.781
With overweight	109 (29.1)	68 (28.6)	41 (29.9)	
With obesity	266 (70.9)	170 (71.4)	96 (70.1)	
BF% measured	39.80 (7.37)	39.78 (7.15)	39.84 (7.77)	*p* = 0.939

BMI = body mass index; BF = body fat.

**Table 2 diagnostics-10-00728-t002:** Correlation between predictors and body fat percentage (BF%) in training and validation samples.

Predictors	Training Sample *n* = 238	Validation Sample *n* = 137
Age	0.069	0.242 **
BMI	0.461 **	0.413 **
Sex	0.635 **	0.560 **

** *p* < 0.001.

**Table 3 diagnostics-10-00728-t003:** Indicators for model performance in the training and validation samples.

Equation	R^2^					
**Training Sample**		**SEE**	**TE**	**Slope**	**Intercept**	***p*-Value ***
Model 1	0.718					
Model 2	0.709	3.81	3.84	1.010	0.179	0.402
**Validation sample ^¥^**		3.88	3.85	0.999	0	0.981
Model 1	0.561					
Model 2	0.588	4.79	4.82	0.894	4.60	0.060

* *t*-test for slope not different from 1. ^¥^ R^2^ is for the regression of BF% on predicted BF%. SEE = standard error of the estimate. TE = total error.

**Table 4 diagnostics-10-00728-t004:** Agreement analysis in training (*n* = 238) and validation (*n* = 137) samples.

	Measured BF%	BF% ^1^	Mean Difference (Bias)	95% CI of Bias (Precision)	Pearson’s Correlation	% Bias	Minimum % Bias	Maximum % Bias	Absolute Percent Error	N (%) Underprediction ^2^	N (%) Accurate Prediction ^3^	N (%) OverPrediction ^4^	% Error	Upper LoA ^5^	Lower LoA	Effect Size	*p*−Value ^6^
Training sample	39.78 ± 7.15																
Model 1		39.20 ± 5.99	0.584 ± 3.81	0.098; 1.07	0.85	−0.53 ± 9.88	−22.92	31.27	7.91 ± 5.93	18.1	67.6	14.3	38.04	8.04	−6.87	0.153	0.019
Model 2		39.80 ± 6.02	−0.017 ± 3.86	−0.51; 0.48	0.84	1.04 ± 10.10	−21.53	32.25	8.16 ± 6.01	17.2	69.3	13.4	38.00	7.54	−7.58	0.004	0.946
Validation sample	39.67 ± 7.21																
Model 1		39.24 ± 6.04	0.425 ± 4.82	−0.39; 1.24	0.75	−0.44 ± 4.82	−33.22	67.16	10.11 ± 9.15	19.7	59.1	21.2	47.62	9.87	−9.02	0.088	0.303
Model 2		39.70 ± 6.15	−0.028 ± 4.67	−0.82; 0.76	0.77	1.49 ± 13.14	−30.46	64.95	9.85 ± 8.78	17.5	60.6	21.9	46.13	9.12	−9.18	0.006	0.943

^1^ BF% = predicted body fat percentage. ^2^ Proportion of people with difference from measured BF% < 10% indicating underprediction. ^3^ Proportion of people with difference from measured BF% between 90% and 110% indicating accurate prediction. ^4^ Proportion of people with difference from measured BF% >1 0% indicating overprediction. ^5^ LoA = limits of agreement. ^6^
*p*-value for paired sample *t*-test.

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
