# Peer review of "Development of an Easy-to-Use Prediction Equation for Body Fat Percentage Based on BMI in Overweight and Obese Lebanese Adults"

_diagnostics, 2020, doi:10.3390/diagnostics10090728_

Round 1

Reviewer 1 Report

The study aimed to develop new gender-specific prediction equations to estimate percentage body fat (%BF) for Lebanese overweight and obese adults using a bioelectrical impedance analysis (BIA) as a reference technique. Using 238 adults to develop equations and 137 adults as an evaluation group, prediction equations with approximately ±8% variability were derived.

The study is well designed and the aim of the study is also clear. Considering a lack of %BF prediction equations for Lebanese populations, particularly for overweight and obese individuals, the equations derived from the present study will have practical benefits to Lebanese in obesity screening. However, there are some issues with the present study as stated below:

- A major concern of the present study was the fact that the equations were derived using BIA as a reference body composition assessment technique. BIA cannot be considered as a gold standard for body composition assessments and while the equation has acceptable variability, this does not mean the estimated value from the prediction equations are accurate. It would be better if the authors utilized other methods such as total body water or dual energy x-ra absorptiometry to develop prediction equations. The authors may explain a reason for choosing BIA as a reference technique for the present study.

- It is not clear if the BIA device used in the study was a single-frequency or multi-frequency device. Since a multi-frequency BIA has higher accuracy and precision compared with a single-frequency device, it is important to provide more detail about the device used.

- In addition, if the BIA device used in the study only provide %BF values and not total body water (TBW) or impedance values, then the authors must acknowledge that the BIA itself uses a population-specific prediction equation which may not be appropriate for Lebanese overweight and obese individuals.

- The study used smaller number of males compared with females, approximately 1/3 in both training and validation samples. Please clarify appropriateness of the sample size.

- p2, line 89. Please change “bio-impedance” to “bioelectrical impedance analysis”.

- p4, line 139. It appease the formulae provided did not appear in a manuscript properly. Please check.

- Figure 1. It is not clear what “0” and “1” for sex indicate.

Author Response

The study aimed to develop new gender-specific prediction equations to estimate percentage body fat (%BF) for Lebanese overweight and obese adults using a bioelectrical impedance analysis (BIA) as a reference technique. Using 238 adults to develop equations and 137 adults as an evaluation group, prediction equations with approximately ±8% variability were derived. The study is well designed and the aim of the study is also clear. Considering a lack of %BF prediction equations for Lebanese populations, particularly for overweight and obese individuals, the equations derived from the present study will have practical benefits to Lebanese in obesity screening.

Response: We thank the reviewer for the appreciation and for the valuable comments.

However, there are some issues with the present study as stated below:

  • A major concern of the present study was the fact that the equations were derived using BIA as a reference body composition assessment technique. BIA cannot be considered as a gold standard for body composition assessments and while the equation has acceptable variability, this does not mean the estimated value from the prediction equations are accurate. It would be better if the authors utilized other methods such as total body water or dual energy x-ray absorptiometry to develop prediction equations. The authors may explain a reason for choosing BIA as a reference technique for the present study.

Response: We completely agree with the reviewer. However, honestly this is the available method for evaluating body composition in our research lab and clinical practice. We highlighted clearly the raised issue in the limitations in the Discussion section (Lines 290-295).

  • It is not clear if the BIA device used in the study was a single-frequency or multi-frequency device. Since a multi-frequency BIA has higher accuracy and precision compared with a single-frequency device, it is important to provide more detail about the device used.

Response: We specified in the Method section that the BIA that we used in our study is a multi-frequency device (Line 110). Moreover we provided details about the device (Lines 111-112 and 115-116) and regard its accurate measurement and agreement with DXA in healthy adults and its validation in overweight and obese individuals in several clinical settings (Lines 121-126).

  • In addition, if the BIA device used in the study only provide %BF values and not total body water (TBW) or impedance values, then the authors must acknowledge that the BIA itself uses a population-specific prediction equation which may not be appropriate for Lebanese overweight and obese individuals.

Response: The BIA device used in our study provide the total body water (TBW) and also the impedance values. This information has been included in the Method section (Lines    115-116).

  • The study used smaller number of males compared with females, approximately 1/3 in both training and validation samples. Please clarify appropriateness of the sample size.

Response: This proportion is representative of the usual weight-management seeking treatment samples accordingly we added a statement in the Discussion section (Lines 283-284) and suitable reference.

  • P2, line 89. Please change “bio-impedance” to “bioelectrical impedance analysis”.

Response: Changed in BIA (Line 95).

  • P4, line 139. It appease the formulae provided did not appear in a manuscript properly. Please check.                                                                        

Response: Changed and now the formulae appear clearly (Line 148).

  • Figure 1. It is not clear what “0” and “1” for sex indicate.

Response: Clarified “0” and “1”, now it appears as males and females (Figure 1).

Reviewer 2 Report

INTRO section

Lines: 46-47. Yes, but at the population level. Lack of concordance is reported at the individual level, though, doi: 10.1371/journal.pone.0200465.

Lines: 48-49. This indeed holds true for the advanced and much extended models. Simple models which may be used in individual clinical practice are much more affordable, which is also well-worth highlighting.

METHODS section

Randomisation was used for the training and validation samples, but how did the Authors ensure that the sample was representative? Please provide more details in this regard. Is Lebanese population characterised by women being three times more numerous than men?

RESULTS section

Line 188. Figure 1, the legend. What 0 and 1 are supposed to mean, respectively? I guess that following the distribution of BF%, women are represented by the blue colour. But this should never be subject to guesswork! The unit of measurement for BMI is incorrect (Kg/m2), as it should read(Kg/m2). In the first two sub-charts the scale of the X-axis is reduced (i.e. it starts off from 10). Perhaps the reader should be advised at this juncture this has actually been done, and also why. The same applies to the Bland Altman plots.

DISCUSSION section

Much too general. I understand the Authors were the first ones to have done something, but they did it for a reason. So, we are looking here at an evolution, not a revolution. There is a vast body of literature on the subject that is worth citing, with a view to providing a rationale for the specific objective pursued by the Authors.

Here is precisely where I seem to be at a loss. Why do the Authors need info on BF%, when they are already in receipt of the BMI values? To assess obesity? Well, not really, as all the study subjects were overweight or obese, which may also be simply defined in terms of the private practices, as already referenced further above. But also because BF% is not recommended for obesity assessment. This is addressed in some detail here, doi:10.4065/mcp.2011.0097. Maybe for CVDs risk assessment, then?

Extensive literature indicates this particular benefit stemming from BF%:  doi.org/10.3945/ajcn.110.001867or doi.org/10.1038/s41598-020-68265-y.Without the awareness of the cut-off points for BF%, usefulness of this measure in predicting CVDs risk is pretty much on a par with knowing the BMI value in assessing obesity without the command of the actual cut-off points for each respective category of this indicator. It is also hard to determine the cut-off points for BF%, if the population under study is characterised by being overweight or obese, only.

I understand that not all the objectives may be addressed within a single study, but I do believe that the DISCUSSION section (and perhaps also the CONCLUSIONS section) are precisely the places where such concerns should be addressed at some length.

Having said this, I also subscribe to the view that the Authors should ensure their review of literature on the subject is always brought as much up to date as practically possible (only a single reference -  no 19 - pertains to 2020), and cover the publications of appreciable impact on the domain they are addressing, notably by the following authors, especially that one of them offers a recently published meta-analysis, simply a "must-have" to be taken into account by any diligent investigators:

  1. Rose R. et al. 2020
  2. Macek P. et al. 2020
  3. Ashwell M. et al. 2016

Finally, I must admit that I am rather impressed with the Authors' methodological prowess in handling the BF% equations.

Author Response

INTRO section

Lines: 46-47. Yes, but at the population level. Lack of concordance is reported at the individual level, though, doi: 10.1371/journal.pone.0200465.

Response: Done as suggested. A statement has been added in the Introduction section and the suggested reference (Lines 49-51).

Lines: 48-49. This indeed holds true for the advanced and much extended models. Simple models which may be used in individual clinical practice are much more affordable, which is also well-worth highlighting.

Response: Done as suggested, and a statement has been added (Lines 53-54).

METHODS section

Randomisation was used for the training and validation samples, but how did the Authors ensure that the sample was representative? Please provide more details in this regard. Is Lebanese population characterised by women being three times more numerous than men?

Response: This proportion is representative of the usual weight-management seeking treatment samples accordingly we added a statement in the Discussion section and suitable reference (Lines 283-284).

RESULTS section

Line 188. Figure 1, the legend. What 0 and 1 are supposed to mean, respectively? I guess that following the distribution of BF%, women are represented by the blue colour. But this should never be subject to guesswork! The unit of measurement for BMI is incorrect (Kg/m2), as it should read (Kg/m2). In the first two sub-charts the scale of the X-axis is reduced (i.e. it starts off from 10). Perhaps the reader should be advised at this juncture this has actually been done, and also why. The same applies to the Bland Altman plots.

Response: All changes have been conducted exactly as suggested. We apologise for the misunderstanding.

DISCUSSION section

Much too general. I understand the Authors were the first ones to have done something, but they did it for a reason. So, we are looking here at an evolution, not a revolution. There is a vast body of literature on the subject that is worth citing, with a view to providing a rationale for the specific objective pursued by the Authors. Here is precisely where I seem to be at a loss. Why do the Authors need info on BF%, when they are already in receipt of the BMI values? To assess obesity? Well, not really, as all the study subjects were overweight or obese, which may also be simply defined in terms of the private practices, as already referenced further above. But also because BF% is not recommended for obesity assessment. This is addressed in some detail here, doi:10.4065/mcp.2011.0097. Maybe for CVDs risk assessment, then?

Response: A short paragraph in the Introduction section has been added that explain the importance of BF assessment and its superiority with respect to BMI, especially in terms of cardiovascular diseases and mortality, as well as suitable references to support it has been added (Lines 32-39).

Moreover the same concept from a different prospective has been reported in the Discussion section (Lines 10-19) to support the aim of our study for the assessment of BF% and not to rely simply on BMI, in line with what has been raised above by the reviewer (Lines 258-267). In addition to suitable references have been added.

Extensive literature indicates this particular benefit stemming from BF%: doi.org/10.3945/ajcn.110.001867or doi.org/10.1038/s41598-020-68265-y. Without the awareness of the cut-off points for BF%, usefulness of this measure in predicting CVDs risk is pretty much on a par with knowing the BMI value in assessing obesity without the command of the actual cut-off points for each respective category of this indicator. It is also hard to determine the cut-off points for BF%, if the population under study is characterised by being overweight or obese, only. I understand that not all the objectives may be addressed within a single study, but I do believe that the DISCUSSION section (and perhaps also the CONCLUSIONS section) are precisely the places where such concerns should be addressed at some length.

Response: In the Discussion section we included the issue raised by the reviewer regard the importance of the establishment of cut-off points for BF% (Lines 305-308), as well as suitable references to support has been added.

Having said this, I also subscribe to the view that the Authors should ensure their review of literature on the subject is always brought as much up to date as practically possible (only a single reference - no 19 - pertains to 2020), and cover the publications of appreciable impact on the domain they are addressing, notably by the following authors, especially that one of them offers a recently published meta-analysis, simply a "must-have" to be taken into account by any diligent investigators:

  1. Rose R. et al. 2020
  2. Macek et al. 2020
  3. Ashwell et al. 2016

Response: Done as suggested, and we did our best to bring our literature review as much up to date.

Finally, I must admit that I am rather impressed with the Authors' methodological process in handling the BF% equations.

Response: We would like thank the reviewer for the excellent comments that for sure led to improvement in our manuscript.